# Analysis of Oral and Gut Microbiome Composition and Its Impact in Patients with Oral Squamous Cell Carcinoma

**DOI:** 10.3390/ijms25116077

**Published:** 2024-05-31

**Authors:** Kensaku Matsui, Ryouji Tani, Sachiko Yamasaki, Nanako Ito, Atsuko Hamada, Tomoaki Shintani, Takeshi Otomo, Koichiro Tokumaru, Souichi Yanamoto, Tetsuji Okamoto

**Affiliations:** 1Department of Molecular Oral Medicine and Maxillofacial Surgery, Graduate School of Biomedical and Health Science, Hiroshima University, Hiroshima 734-8553, Japan; k-matsui@hph.pref.hiroshima.jp (K.M.); tetsuok@hiroshima-u.ac.jp (T.O.); 2Department of Oral and Maxillofacial Surgery, Hiroshima University Hospital, Hiroshima 734-8553, Japan; nanainc7@hiroshima-u.ac.jp; 3Department of Oral Oncology, Graduate School of Biomedical and Health Science, Hiroshima University, Hiroshima 734-8553, Japan; sayamasaki@hiroshima-u.ac.jp (S.Y.); hamaco@hiroshima-u.ac.jp (A.H.); syana@hiroshima-u.ac.jp (S.Y.); 4Center of Oral Clinical Examination, Hiroshima University Hospital, Hiroshima 734-8553, Japan; tshintan@hiroshima-u.ac.jp; 5NIHON KEFIA Co., Ltd., 13-16, Asahicho, Fujisawa-shi 251-0054, Japan; totom@kefir.co.jp (T.O.); ktokum@kefir.co.jp (K.T.); 6School of Medical Sciences, University of East Asia, Shimonoseki 751-8503, Japan

**Keywords:** gut microbiome, oral microbiome, oral squamous cell carcinoma, 16S rRNA, programmed cell death-ligand 1 (PD-L1)

## Abstract

The impact of gut and oral microbiota on the clinical outcomes of patients with oral squamous cell carcinoma (OSCC) is unknown. We compared the bacterial composition of dental plaque and feces between patients with OSCC and healthy controls (HCs). Fecal and dental plaque samples were collected from 7 HCs and 18 patients with OSCC before treatment initiation. Terminal restriction fragment-length polymorphism analysis of 16S rRNA genes was performed. Differences in bacterial diversity between the HC and OSCC groups were examined. We compared the occupancy of each bacterial species in samples taken from patients with OSCC and HCs and analyzed the correlation between PD-L1 expression in the tumor specimens and the occupancy of each bacterial species. The gut and oral microbiota of patients with OSCC were more varied than those of HCs. *Porphyromonas* and *Prevotella* were significantly more abundant in patients with OSCC than in HCs. The abundance of *Clostridium* subcluster XIVa in the gut microbiota of the PD-L1-positive group was significantly greater than that in the PD-L1-negative group. The oral and gut microbiomes of patients with OSCC were in a state of dysbiosis. Our results suggest the possibility of new cancer therapies targeting these disease-specific microbiomes using probiotics and synbiotics.

## 1. Introduction

Approximately 1000 species of gut bacteria, up to 100 trillion in total, coexist in the human intestinal tract, and this microbiome has been considered a substantial organ [1]. Dysbiosis is involved in the development of various diseases, such as inflammatory bowel disease, cancer, obesity, and diabetes [2,3,4,5,6,7,8,9]. Furthermore, the oral commensal microbiome is not only involved in infectious diseases, such as periodontitis and dental caries but may also be associated with gut microbiome and systemic diseases, although the details of this relationship remain unclear [10,11,12,13,14,15].

Recently, a close relationship between the antitumor effects of various treatments and the gut microbiomes of patients with cancer was reported [16,17]. As novel cancer immunotherapies, therapeutic monoclonal antibodies targeting programmed cell death protein 1 (PD-1) and programmed death-ligand 1 (PD-L1) have been developed and are already being used in the clinic. The PD-1 molecule was isolated and identified as a gene encoding a type I membrane protein whose expression is induced by the stimulation of T-cell death [18]. Subsequently, the growth of murine melanoma transplanted into PD-1 knockout mice or transplanted mouse tumors was inhibited by antibodies that block the PD-1/PD-L1 pathway, indicating that this pathway plays a central role in cancer immune escape [19,20]. In one study, mice were classified into two groups based on different proportions of bacteria in their gut microbiomes. The antitumor effects of anti-PD-L1 antibodies were examined and found to be significantly greater in mice with a *Bifidobacterium*-dominant gut microbiome than in the other groups. This suggests that the antitumor effects of checkpoint inhibitors are affected by the gut microbiome [21]. Additionally, the antitumor effects of platinum-based anticancer agents are reduced in germ-free mice compared with those in conventional mice [22]. Therefore, the effects of platinum-based anticancer agents might be influenced by the microbiome of the organism. We have previously shown that patients with oral cancer can achieve improved liver function by intaking Kefir (NIHON KEFIA Co., Ltd., Fujisawa, Japan), a fermented food, during cancer treatment, suggesting that gut microbiota have various effects on pathological conditions [23,24].

Metabolites produced by gut bacteria may affect the clinical pathophysiology of patients with cancer and influence the antitumor effects of cancer treatment. In particular, short-chain fatty acids, which are metabolites of intestinal bacteria, are not only utilized as an energy source in host cells but also act as an energy regulatory mechanism via short-chain fatty acid receptors, such as G-protein-coupled receptor (GPR) 41 and GPR43 in adipocytes [25,26,27], inhibiting histone deacetylase in lymphoid cells and inducing differentiation into regulatory T cells by epigenomic changes [28]. Therefore, short-chain fatty acids produced by gut bacteria influence the maintenance of homeostasis and immune regulation in living organisms. However, the impact of gut microbiota on oral squamous cell carcinoma (OSCC) remains unknown.

Oral microflora comprises 500–700 bacterial species, which form characteristic bacterial populations in the following sites: dental surfaces, periodontal pockets, saliva, tongue surfaces, and buccal mucosa. Oral microflora has been associated with a number of conditions, including dental caries, periodontal disease, other mucosal diseases, and systemic diseases [10,11,12,13,14,15]. Recently, it has been reported that alterations in the microbiota are associated with the development of human cancer and particularly that some specific bacterial strains are strongly associated with the development of oral cancer [29]. Yang et al. conducted an analysis of saliva samples from OSCC patients, which revealed an imbalance in oral taxa, with a relative abundance of *Bacteroidetes and Firmicutes* in three different oral cancer groups clustered according to the cancer mutation status. Moreover, the analysis revealed significant variability in microbial diversity across the three patient groups [30].

The objective of this study was to examine the null hypothesis that there is no difference in the bacterial composition of the oral and intestinal microbiota of healthy subjects and oral cancer patients. Terminal restriction fragment-length polymorphism (T-RFLP) analysis was conducted on the gut and oral microbiomes of patients with OSCC to investigate the impact of these microbiomes on cancer immunity and clinical pathogenesis.

## 2. Results

### 2.1. Baseline Characteristics

Of the 18 patients with OSCC, 10 and 8 were men and women, respectively, with a mean age of 65.0 years. Of the seven HCs, four were men, and three were women, with a mean age of 60.7 years. N0, N1, N2, and N3 cervical lymph node metastases were observed in one, nine, three, and five patients, respectively. Overall, one, eight, four, and five patients were in Stages I, II, III, and IV, respectively (Table 1).

### 2.2. Comparison of Diversity between the OSCC and HC Groups in Gut and Oral Microbiomes

The bacterial species detected through T-RFLP are listed in Table 2. Principal coordinate analysis revealed significant clustering of gut and oral microbiomes in the OSCC and HC groups (Figure 1).

In terms of the Euclidean distances within the HC group (HC-HC), between the HC and OSCC groups (HC-OSCC), and within the OSCC group (OSCC-OSCC), no differences were found in any of the comparisons of the gut microbiome (Figure 2A). When comparing distances in the oral microbiome, significant differences were found in all group comparisons when the restriction enzyme *Hha* I was used (*p* < 0.01, Figure 2B). On the other hand, those of HC-OSCC and OSCC-OSCC were significantly greater than those of HC-HC when the restriction enzyme *MSP* I was used (*p* < 0.01, Figure 2C).

### 2.3. Comparison of Bacterial Abundances between the Gut and Oral Microbiomes of the OSCC and HC Groups

In the gut microbiome, there was a significant difference in abundance between the OSCC and HC groups, with only the other groups showing significant differences (*p* < 0.05, Figure 3A). Digestion with the restriction enzyme *Hha* I showed that *Streptococcus* and *Eubacterium* were significantly less abundant, and *Porphyromonas* and *Prevotella* were significantly more abundant in the OSCC group than in the HC group (*p* < 0.05, Figure 3B). Analysis with the restriction enzyme *Msp* I showed that *Streptococcus* was significantly less abundant, and *Eubacterium* and *Parvimonas* were significantly more abundant in the OSCC group than in the HC group (*p* < 0.05, Figure 3C).

### 2.4. Comparison of Clinicopathological Factors between the PD-L1-Positive and -Negative Groups and the PD-1-Positive and -Negative Groups

Immunohistochemical examination of PD-L1 and PD-1 expression in OSCC tissues revealed 5 patients in the PD-L1-positive group, 13 patients in the PD-L1-negative group, 6 patients in the PD-1-positive group, and 12 patients in the PD-1-negative group. No significant differences in clinicopathological factors were found between the PD-L1- or PD-1-positive and -negative groups (Table 3).

### 2.5. Comparison of the Prognosis between the PD-L1-Positive and -Negative Groups and the PD-1-Positive and -Negative Groups

As a distinct patient cohort was used in this study, the patient characteristics are presented in Appendix A. Notably, the OS rate exhibited a significant decrease in the PD-L1-positive group compared to the PD-L1-negative group (*p* < 0.05, Appendix A). No differences were found between the PD-1-positive and PD-1-negative groups (Appendix A).

### 2.6. Comparison of Bacterial Occupancy between the PD-L1-Positive and PD-L1-Negative Groups

A comparison of bacterial abundance between the PD-L1-positive and PD-L1-negative groups revealed that the abundance of *Clostridium* subcluster XIVa in the gut microbiota was significantly greater in the PD-L1-positive group than in the PD-L1-negative group (*p* < 0.05, Figure 4). There were no significant differences in the occupancy of other bacteria between the two groups. Similarly, no differences were found in the oral microbiome between the PD-L1-positive and PD-L1-negative groups (*p* < 0.05, Appendix A).

## 3. Discussion

Recently, dysbiosis in the gut microbiome has been reported to be involved in the development and progression of numerous systemic diseases [31]. In the present study, the gut and oral microbiomes of the OSCC and HC groups were identified. PCA and cluster analyses revealed that the microbiome of the OSCC group was different from that of the HC group. The abundance of periodontal pathogens, such as *Porphyromonas* and *Prevotella*, in the oral microbiome was significantly greater in patients with oral cancer than in healthy individuals, suggesting that oral indigenous bacteria, such as *Porphyromonas*, might be involved in the development and progression of cancer [32]. Recently, new findings have been presented regarding oral and pharyngeal cancers related to intestinal microbiota. In gut microbiota, *Prevotellaceae* was a risk factor for oral and oropharyngeal cancer in the European population but a protective factor in the North American population. Based on our results, *Prevotellaceae* are a risk factor in the oral microflora, which differs from the findings of a previous report [33]. It is unclear whether this dysbiosis is involved in the development or progression of oral cancer. However, the improvement in disease-specific dysbiosis can result in a therapeutic or preventive effect, as exemplified by treatment initiation of pseudomembranous colitis by fecal transplantation [34,35]. Although the bacterial flora involved in the development and progression of OSCC has been previously reported, these findings differ from the results of this study. An association between cancer and dysbiosis has been reported previously [21], and in oral cancer, an association between periodontitis and OSCC [36,37,38,39] and *Porphyromonas gingivalis* infection might increase the risk of oral and gastrointestinal cancers [40]. *Porphyromonas gingivalis*, a member of the *Porphyromonas* group, was found to be significantly more prevalent in the OSSC group than in the HC group. This bacterium has been reported to interfere with numerous host responses to oral cancer. This bacterium activates Toll-like receptors 2 and 4, resulting in the release of pro-inflammatory cytokines, such as interleukin-8 (IL-8) [41]. Furthermore, gingipain, a proteolytic enzyme produced by *Porphyromonas gingivalis*, has been demonstrated to cleave matrix metalloproteinase 9 (MMP9) and activate it to its mature form. This process results in the degradation of basement membranes and contributes to the metastasis of OSCC [42].

In addition, one study reported that significantly more *Fusobacterium nucleatum* was detected in esophageal cancer tissues than in normal tissues [43].

As mentioned above, past reports have typically focused on the oral flora in patients with oral cancer; however, few studies have investigated the gut microbiome and immune response in oral cancer patients [44]. Therefore, we examined and compared the bacterial flora between groups with and without PL-L1 expression in tumors. In particular, the mechanism of immune evasion in the clinical pathogenesis of oral cancer in the microenvironment remains unclear. We first compared OS rates in the PD-L1- and PD-1-positive and -negative groups and found that the OS rate in the PD-L1-positive group was significantly lower than that in the PD-L1-negative group. Cho et al. and Lin et al. previously reported that patients exhibiting high PD-L1 expression tend to experience lower survival rates compared to those with low PD-L1 expression, suggesting that high PD-L1 expression might serve as a poor prognostic factor [45,46]. These findings align well with our results. Subsequently, we investigated the correlation between PD-L1 expression and bacterial abundance within each microbiome group, focusing on both gut and oral microbiomes. The results show that the abundance of *Clostridium* cluster XIVa was significantly greater in the PD-L1-positive group than in the PD-L1-negative group in the gut microbiome, although no significant correlation was detected in the oral microbiome. Butyrate, a short-chain fatty acid produced by anaerobic metabolism in the gastrointestinal tract by *Clostridiales* species, such as those of *Clostridium* cluster XIVa, has been reported to promote the differentiation of regulatory T cells, which play an important role in suppressing immune responses [28].

Short-chain fatty acids produced by gut bacteria are reportedly involved in diseases such as chronic kidney disease, diabetes, and colorectal cancer, indicating that they have diverse functions as biological modifiers [47,48,49]. We hypothesize that butyrate produced by gut bacteria, such as clostridia, might affect PD-L1 expression in tumor cells in the cancer microenvironment and could be involved in immune regulatory mechanisms via PD-L1/PD-1 signaling. In addition, butyrate, which is continuously supplied to cancer lesions via the oral microbiome, might have a similar effect on the cancer microenvironment. We investigated the impact of short-chain fatty acids on PD-L1 mRNA and protein expression in A431 cells in vitro and found that succinate, propionate, and butyrate elevated both the PD-L1 gene and protein expression in A431 cells suggesting involvement of these short-chain fatty acids in the cancer microenvironment. The observed rise in *Clostridium* cluster XIVa, a butyrate-producing bacterium, potentially correlates with increased PD-L1 expression, consequently contributing to immune response suppression.

This study has several limitations. First, as this was a single-center, case-control study with a small sample size, regional bias might have occurred. Second, factors such as smoking history, alcohol consumption history, and systemic diseases were not considered. In the treatment of oral cancer, the environment of the oral cavity and gastrointestinal tract changes dramatically before and after treatment, including the administration of antimicrobial agents to prevent postoperative infections, mucosal damage, and myelosuppression due to chemotherapy and radiation therapy, changes in the diet, and the addition of gastrostomy. To elucidate the relationship between these dynamic changes and the microbiome and the relationship between the presence or absence of recurrence or metastasis and prognosis or the effects of cancer treatment, both short-term studies during cancer treatment and long-term prospective studies after treatment are needed.

Nevertheless, this study underscores the significance of gut and oral microbiomes in the development and pathogenesis of oral cancer, proposing potential targets for novel cancer therapies aimed at microbiomes. Moreover, we are the first to unveil the association between gut microbiota in patients with OSCC and the expression of PD-L1, a pivotal immune checkpoint-related molecule. The combined use of probiotics and synbiotics, which improve dysbiosis of the gut microbiome, in the treatment of oral cancer is expected to improve therapeutic efficacy and prognosis and reduce the incidence of secondary cancers.

## 4. Materials and Methods

### 4.1. Participants

Between November 2016 and September 2017, 18 patients with OSCC and 7 healthy controls (HCs) who underwent treatment at the clinical department of Oral and Maxillofacial Surgery, Hiroshima University Hospital, were included and registered in this case-control study. The inclusion criteria were the availability of fecal matter, dental plaque, archived biopsy specimens before treatment and clinical data, pathological diagnosis of OSCC, and 6 months or more of follow-up care for living participants. The exclusion criteria were a history of surgery on the primary tumor, chemotherapy, and radiotherapy. Resected formalin-fixed, paraffin-embedded (FFPE) tissue samples were obtained before treatment. Overall survival (OS) was calculated from the date of surgery. The data collected from the participants’ medical charts included age, sex, site of the lesion, treatment details, and disease staging according to the TNM classification of the International Union for Cancer Control, 8th ed. PD-L1 and PD-1 expression was examined immunohistochemically using cohorts of different durations between January 2001 and September 2013, with 72 available biopsy specimens.

### 4.2. Collection of Fecal Matter and Dental Plaque

Fecal matter was collected from HCs and patients with OSCC before treatment using a stool collection kit (containing 100 mM Tris-HCl [pH 9], 40 mM ethylenediaminetetraacetic acid [EDTA], 4 M guanidine thiocyanate, and 0.001% bromotimol; TechnoSuruga Laboratory Co., Shizuoka, Japan) and stored at 4 °C. Dental plaque was collected from the tooth surface using sterile cotton swabs and stored at −80 °C. The collected samples were submitted to the TechnoSuruga Laboratory for T-RFLP analysis as described subsequently. Bacteria from fecal samples were defined as the gut microbiome, and those from dental plaque were defined as the oral microbiome.

### 4.3. DNA Extraction

DNA extraction from fecal samples was performed according to the method described by Takahashi et al. [50]. Each sample was resuspended in 4 M guanidium thiocyanate, 100 mM Tris-HCl (pH 9.0), and 40 mM EDTA and homogenized with zirconia beads using a FastPrep FP100A device (MP Biomedicals, Santa Ana, CA, USA). The Magtration System 12GC and GC series MagDEA DNA 200 (Precision System Science, Chiba, Japan) were used to extract DNA from the bead-treated suspensions. DNA concentrations were estimated by spectrophotometry using an ND-1000 instrument (NanoDrop Technologies, Wilmington, DE, USA), and the final concentration of the DNA samples was adjusted to 10 ng/µL.

DNA extraction from dental plaque was performed using the MORA-EXTRACT kit (Kyokuto Pharmaceutical Industrial Co., Ltd., Tokyo, Japan), and the cotton swabs used for the collection were immersed in 200 µL of lysis buffer. All liquid was transferred to a bead-filled tube and kept at 70 °C for 10 min to promote lysis. A disruptor generator (Scientific Industries, Bohemia, NY, USA) was used to disrupt the sample for 2 min. After crushing, 200 µL of sodium dodecyl sulfate solution was added to the tube, and the lysis was accelerated at 70 °C for 10 min.

### 4.4. T-RFLP Analysis

PCR amplification of 16S ribosomal RNA (rRNA) from fecal samples was performed according to the method reported by Nagashima et al. [51,52]. The primer set included 6′-carboxyfluorescein (6-FAM)-labeled 516F primers (5′-TGCCAGCAGCCGCGCGGTA-3′; *Escherichia coli* positions 516–532) and 1510R primers (5′-GGTTACCTTGTTACGACTT-3′; *E. coli* positions 1510–1492). DNA was purified using a MultiScreen PCR96 Filter Plate (Millipore, Billerica, MA, USA). A total of 10 U/10 µL of *Bsl* I (5′-CCNNNNN|NNGG-3′, New England BioLabs, Ipswich, MA, USA) was used for digestion at 55 °C for 3 h [32]. The obtained fluorescently labeled terminal restriction fragments (T-RFs) were analyzed using an ABI PRISM 3130xl Gene Analyzer (Applied Biosystems, Waltham, MA, USA), and their lengths and peak areas were determined using GeneMapper 4.0 (Applied Biosystems). The T-RFs were classified into 29 operational taxonomic units (OTUs). OTUs were quantified as the percentage of individual OTUs per total OTU area, which was expressed as the percent area under the curve (%AUC). For each taxonomic unit, bacteria were predicted, and the corresponding OTUs were identified according to the reference human fecal microbiome based on T-RFLP profiling, (https://www.tecsrg.co.jp/, accessed on 20 January 2018).

PCR amplification of 16S rRNA of dental plaque specimens was performed as reported by Sakamoto et al. [53]. The primers 6-FAM-labeled 27F (5′-AGAGAGTTTGATCCTGGCTCAG-3′) and 1492R (5′-GGTTACCTTGTTACGACTT-3′) were employed. The PCR products were purified using the PEG precipitation method [54]. As reported by Sakamoto et al., 20 U/10 µL of *Hha* I (5-GCG|C-3, Takara Bio, Kusatsu, Japan) and *Msp* I (5′-C|CGG-3′, Takara Bio) were used for digestion at 37 °C for 3 h [51]. Both dental plaque and feces were subjected to T-RF analysis, and each taxonomic unit was estimated and identified (Figure 5).

### 4.5. Immunohistochemical Analysis

Sections (5 μm) were prepared from 10% formalin-fixed paraffin-embedded specimens, deparaffinized with xylene and ethanol, and treated with 10 mM citrate, pH 4.0, or 1 mM EDTA, pH 8.0, for 15 min at 96 °C in the dark to stimulate the antigen. Next, we added 0.3% hydrogen peroxidase-containing methanol for 10 min at room temperature under light shielding to block endogenous peroxidase activity. Subsequently, the samples were washed with TBST and incubated with the primary antibody at 4 °C overnight. When using rabbit anti-PD-L1 polyclonal antibody (1:200; Proteintech Group Inc., Tokyo, Japan) as the primary antibody, the sections were treated with EDTA and reacted with a mouse anti-PD-1 monoclonal antibody (1:200; EH33, Cell Signaling Technology Japan, K.K., Tokyo, Japan). HRP-conjugated dextran-binding anti-rabbit IgG antibody from Dako’s EnVision kit (DAKO A/S, Copenhagen, Denmark) was used as a secondary antibody. The samples were allowed to react at room temperature for 30 min. Coloration was performed with diaminobenzidine, nuclear staining was performed with hematoxylin, and dehydration, penetration, sealing, and speculum examination were subsequently performed, as previously described [55,56].

For PD-L1 expression, three fields of view around the periphery of the tumor invasion in the stained specimen were randomly selected. The expression level was calculated as the product of the color intensity score and the percentage of positive cells: for the intensity score, no staining = 0, weak staining = 1, moderate staining = 2, and strong staining = 3; for the percentage of positive cells, 0% = 0, 1–30% = 1, 30–60% = 2, and 60–100% = 3. The number of PD-1-positive cells was measured in the three fields of view where the density of PD-1-positive cells in the tumor was highest, and the average number of PD-1-positive cells in the three fields was calculated as follows: the top 50% of the average number of PD-1-positive cells in the three fields of view comprised the positive group, and the bottom 50% comprised the negative group.

### 4.6. Statistical Analysis

Continuous variables are expressed as the mean ± standard deviation (SD) or median (interquartile range [IQR]), while discrete variables are expressed as frequencies. The statistical significance of intergroup differences was assessed using the chi-square test (for discrete variables). Comparisons among multiple groups in the Euclidean distance analysis were analyzed using the Kruskal–Wallis test, followed by Steel–Dwass multiple comparison tests. Principal component analysis (PCA) was performed to examine the differences in microbiome structures between the different groups. The Euclidean distance was used as a similarity measure between specimens. The occupancy rate of each bacterium in the OSCC and HC groups was compared using Student’s *t*-test. The abundance of each bacterium between PD-L1-positive and -negative groups and PD-1-positive and PD-1-negative groups was assessed using the Mann–Whitney *U* test. Survival curves were calculated using the Kaplan–Meier method, and significant differences in OS rates between PD-L1-positive and -negative groups and PD-1-positive and -negative groups were determined using the log-rank test. A risk rate of 5% or less was considered indicative of a significant difference. Statistical analyses were conducted using JMP Pro 14 (SAS Institute Inc., Cary, NC, USA).

### 4.7. Ethical Considerations

This retrospective case-control study was approved by the Research Ethics Board of Hiroshima University. The epidemiology-439 protocol was approved on 27 July 2016 and epidemiology-697 on 28 February 2017. The study protocol was posted on our institution’s website. Written informed consent was obtained from all participants.

## 5. Conclusions

The gut and oral microbiomes of patients with OSCC were found to be in a state of dysbiosis. In addition, the abundance of *Clostridium* subcluster XIVa in the gut microbiome was significantly greater in the PD-L1-positive group than in the PD-L1-negative group of patients with OSCC. Short-chain fatty acids, such as butyrate, produced by *Clostridium* might impact therapeutic efficacy by regulating the expression of immune checkpoint molecules in the cancer microenvironment. Our findings suggest the possibility of new cancer therapies targeting these disease-specific microbiomes using probiotics and synbiotics, which can improve dysbiosis of the gut microbiome.

## Figures and Tables

**Figure 1 ijms-25-06077-f001:**
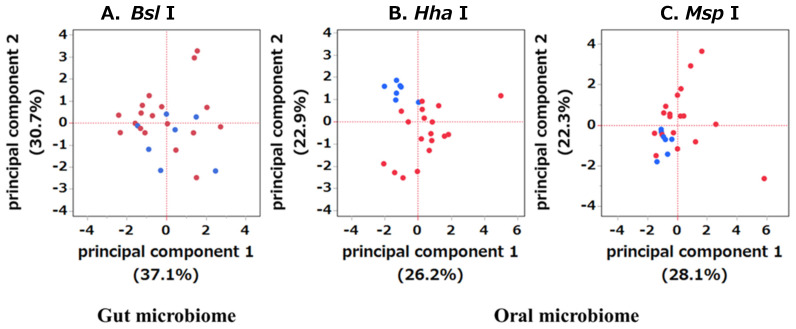
Comparison of principal component analysis (PCA) between the gut and oral microbiomes of the HC and OSCC groups. The microbiome is distinct between stools (**A**) and dental plaque (**B**,**C**). PCA plots depicting the HC (blue) and OSCC (red) groups.

**Figure 2 ijms-25-06077-f002:**
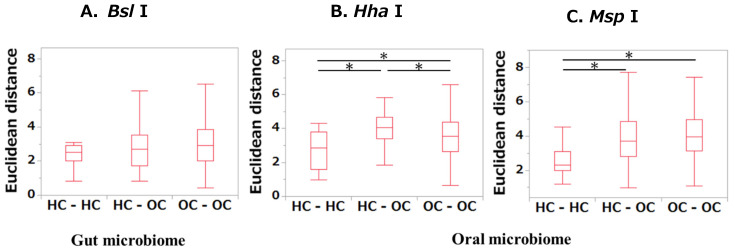
Interparticipant variability among the HC and OSCC groups according to Euclidean distance analyses. For microbiome samples, Euclidean distance was employed to calculate the distance for each sample. Variability between HCs, among OSCCs, and between HCs and OSCC patients was assessed. Comparisons among multiple groups were analyzed using the Kruskal–Wallis test, followed by Steel–Dwass multiple comparison tests. * *p* < 0.05 (statistically significant).

**Figure 3 ijms-25-06077-f003:**
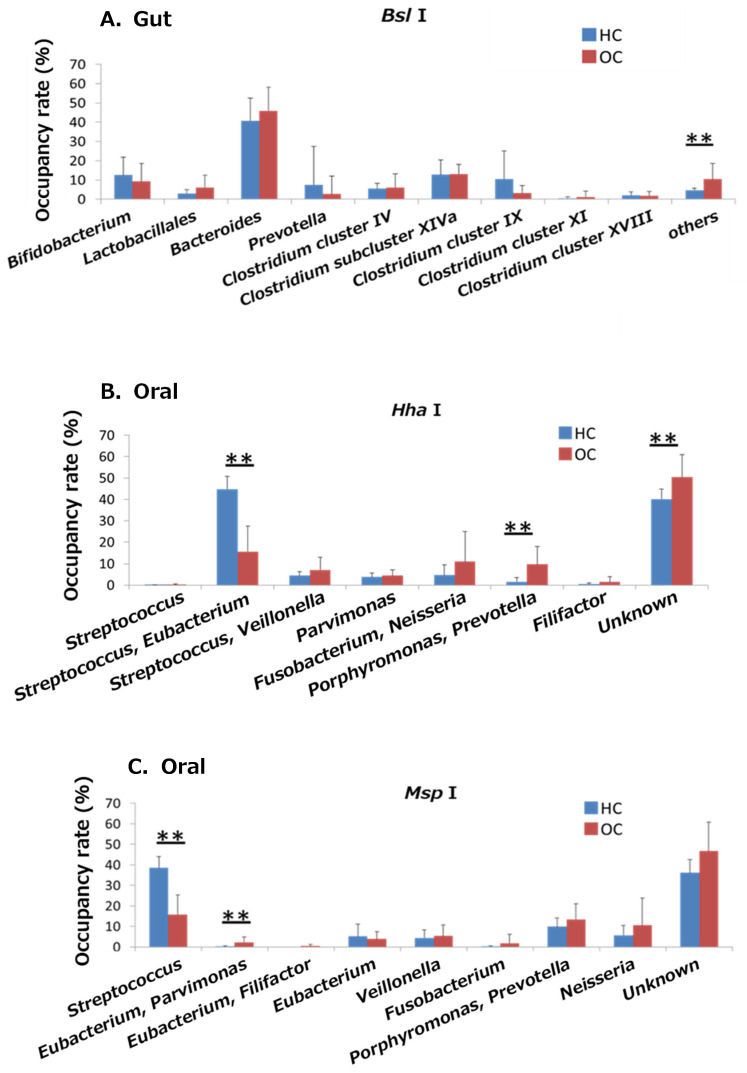
Comparison of the occupancy rates of the gut and oral microbiomes between the HC and OSCC groups. The percentage of bacteria detected in each sample was compared between the OSCC and HC groups. ** *p* < 0.01 (statistically significant).

**Figure 4 ijms-25-06077-f004:**
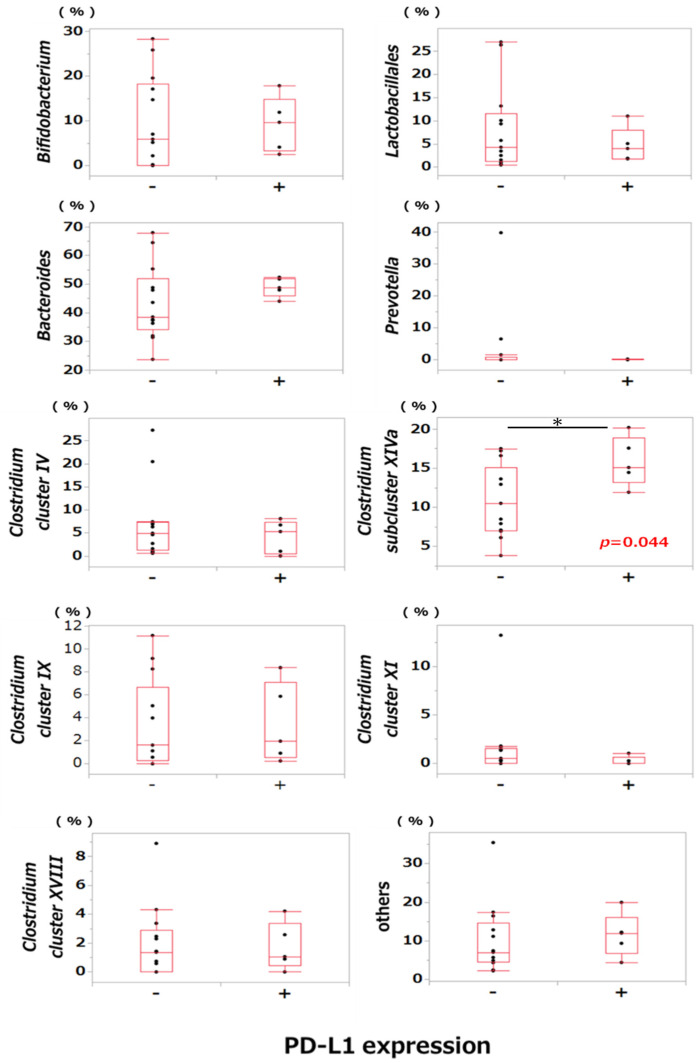
Comparison of the occupancy rates of the gut microbiome between the PD-L1-positive and PD-L1-negative groups of patients with OSCC. Differences in the occupancy rate between the PD-L1-positive and PD-L1-negative groups of patients with OSCC in the gut microbiome were compared. * *p* < 0.05 (statistically significant).

**Figure 5 ijms-25-06077-f005:**
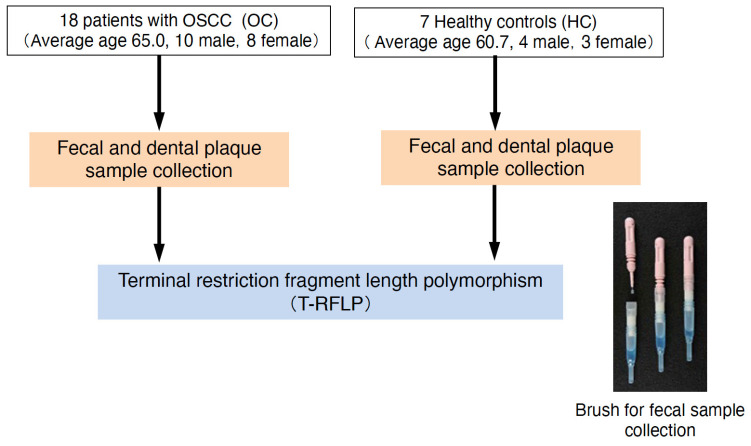
Flowchart of this study.

**Table 1 ijms-25-06077-t001:** Baseline characteristics of patients with oral cancer and healthy controls.

Category		OC (*n* = 18)	HC (*n* = 7)
Sex	(male/female)	10/8	4/3
Age	average (range)	65.0 (32–84)	60.7 (32–94)
T	1	1	
	2	9	
	3	3	
	4	5	
N	negative	14	
	positive	4	
Stage	Ⅰ	1	
	Ⅱ	8	
	Ⅲ	4	
	Ⅳ	5	

HC: healthy controls; OC: patients with oral cancer.

**Table 2 ijms-25-06077-t002:** Bacterial groups detectable by terminal restriction fragment-length polymorphism (T-RFLP).

Gut Microbiome	Oral Microbiome	
<restriction enzyme: *Bsl I*>	<restriction enzyme: *Hha I*>	<restriction enzyme: *Msp I*>
*Bifidobacterium*	*Streptococcus*	*Streptococcus*
*Lactobacillales*	*Streptococcus*, *Eubacterium*	*Eubacterium*, *Parvimonas*
*Bacteroides*	*Streptococcus*, *Veillonella*	*Eubacterium*, *Filifactor*
*Prevotella*	*Parvimonas*	*Eubacterium*
*Clostridium* cluster IV	*Fusobacterium*, *Neisseria*	*Veillonella*
*Clostridium* cluster IX	*Porphyromonas*, *Prevotella*	*Fusobacterium*
*Clostridium* cluster XI	*Filifactor*	*Porphyromonas*, *Prevotella*
*Clostridium* cluster XIVa	others	*Neisseria*
*Clostridium* cluster XVIII		others
others		

**Table 3 ijms-25-06077-t003:** Comparison of clinical pathological factors between PD-L1- and PD-1-positive and -negative groups.

Category		PD-L1		*p*-Value	PD-1		*p*-Value
		Positive (*n* = 5)	Negative (*n* = 13)		Positive (*n* = 6)	Negative (*n* = 12)	
Sex	(male/female)	4/1	5/8	0.291 ^1^	2/4	7/5	0.621 ^1^
Age	median (IQR)	62.4 (40–68)	69.1 (40–84)	0.372 ^2^	71.8 (60–82)	64.9 (40–84)	0.332 ^2^
T	1	0	1	0.411 ^1^	1	0	0.451 ^1^
	2	2	7		2	7	
	3	2	1		1	2	
	4	1	4		2	3	
N	negative	4	10	0.671 ^1^	6	8	0.111 ^1^
	positive	1	3		0	4	
Stage	Ⅰ	0	1	0.61 ^1^	1	0	0.481 ^1^
	Ⅱ	2	6		2	6	
	Ⅲ	2	2		1	6	
	Ⅳ	1	4		2	3	

^1^ Chi-square test. ^2^ Wilcoxon rank-sum test. IQR: interquartile range; PD-1: programmed cell death 1; PD-L1: programmed cell death-ligand 1.

## Data Availability

All the data generated or analyzed during this study are included in this published article.

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
