# Peer review of "Analysis of Oral and Gut Microbiome Composition and Its Impact in Patients with Oral Squamous Cell Carcinoma"

_ijms, 2024, doi:10.3390/ijms25116077_

Round 1

Reviewer 1 Report

Comments and Suggestions for Authors

Dear authors,

Thank you for submitting your valuable work to the journal. The topic of your research is intersting and brings a novel approach to the pathogenesis of the oral cancer, from the perspective of oral biofilm dysbiosis. 

The paper is generally well written and scientfically sound. However, there are some comments I would make in order to increase its scientific accuracy:

- please add a null hypothesis to the objectives of your study

- were the participants subjected to any periodontal assessment prior to the sampling of biofilm? This could have been significantly relevant to study's results

- were inclusion/exclusion criteria applied regarding the oral health status of participating patients? (such as dental implants/removable prostheses or prior periodontal therapy) this could have impacted the results

- if the patients were examined between 2016 and 2017 why has been the manuscript submitted in 2024? this is 7 year time span. please explain

- please provide exact date of ethical approval declaration

- the majority of the used references for the discussions are dated before 2020, an update of existing literature should be performed and included in order to make the results relevant for 2024

We look forward to receiving the revised version of your manuscript. 

Kind regards. 

Comments on the Quality of English Language

Moderate editing

Reviewer 2 Report

Comments and Suggestions for Authors

After reading the article Analysis of the oral and gut microbiome composition and its 2 impact in patients with oral squamous cell carcinoma, whose objective was to compare the bacterial composition of dental plaque and feces between patients with OSCC and healthy controls (HCs), the results I found them interesting, however, I have a suggestion:

1. More needs to be discussed about the microbial dysbiosis that was found in patients with OSCC and HC, that is, how this microbial dysbiosis affects the relationship with patients with OSCC at a molecular level. Furthermore, could it be a consequence of pharmacokinetic effects?

In general it is a well planned and very interesting project.

Comments on the Quality of English Language

Minor editing of English language required

Reviewer 3 Report

Comments and Suggestions for Authors

1. What is the Role of Oral Bacteria in the Development of Oral Squamous Cell Carcinoma?

2. What is the composition of the oral microbiome?

3. What are the mechanisms and Potential Clinical Implications of Oral Microbiome in Oral Squamous Cell Carcinoma?

4. What are the variations in oral microbiota associated with oral cancer?

Round 2

Reviewer 1 Report

Comments and Suggestions for Authors

Dear authors,

Thank you for providing the revised version of your manuscript. My concerns have been properly addressed by the applied changes and the additional explanations. The paper has improved significantly in scientific accuracy. 

I have no further comments.

Kind regards. 

Comments on the Quality of English Language

Moderate check-up.